# Decomposition of Heavy Diesel SCR Urea Fluid Adsorbed in Cu/HZSM-5 SCR Catalysts Studied by FTIR Spectroscopy at Ambient Conditions

Christiaan Tempelman [1,*], Brahim el Arkoubi [1], Jochem Spaan [1], Ronny Slevani [1] and Volkan Degirmenci [2]

[1] Department of Chemical Engineering, Rotterdam Mainport Institute, Rotterdam University of Applied Sciences, Lloydstraat 300, 3024 EA Rotterdam, The Netherlands
[2] School of Engineering, University of Warwick, Coventry CV4 7AL, UK
* Correspondence: c.h.l.tempelman@hr.nl

**Abstract:** A method is presented to study the decomposition of urea deposited on Cu/HZSM-5 SCR catalysts and therewith the ability of the Cu/HZSM-5 SCR catalyst to be regenerated when being overdosed with SCR urea fluids during operation. This straightforward laboratory method could speed up calibration of exhaust gas aftertreatment systems. As an example, the removal of adsorbed urea to the SCR substrate due to dosage malfunction is studied. To study the removal of adsorbed urea on the catalyst substrate, FTIR experiments have been conducted to investigate the state of the catalyst. Besides Cu/HZSM-5 also HZSM-5 and CuOx were studied as model compounds to provide more inside on the processes occurring at the Cu/HZSM-5 surface upon urea decomposition. To simulate exposure of the SCR catalyst to overdosing of the urea solution, samples were impregnated with a 32 wt% urea solution, which correlates to that of commercial heavy duty diesel urea solutions. After impregnation, the samples were heated at various temperatures in the 133–400 °C temperature region, typically the operation window of a SCR catalyst. After heating, the samples were cooled to room temperature and measured in FTIR. The obtained spectra were compared with various literature reports to correlate the observed absorption bands to urea, urea related compounds and decomposition compounds. The concentration of these adsorbed species decreases at increased thermolysis temperature and is no longer visible at temperatures >250 °C. Extended heat treatment at 200 °C revealed only minor loss of adsorbents after 6 h and were still observable in the FTIR spectra after 24 h. Urea derived adsorbents were completely removed when simulating catalyst regeneration under SCR operation conditions under continuous air flow with a humidity of 10% and at elevated temperatures (400 °C).

**Keywords:** Cu/HZSM-5; urea; FTIR; SCR; regeneration

## 1. Introduction

Current regulations restrict the concentrations of $NO_x$ components emitted by the exhaust gas of heavy-duty diesel engines [1,2]. In order to reduce the amount of NO and $NO_2$, the exhaust gas is treated by a Selective Catalytic Reduction (SCR) catalyst [3]. To enable $NO_x$ reduction, urea is added as an aqueous solution in the exhaust gas stream [4]. During injection of urea in the hot exhaust gas stream, evaporation of the water solvent and decomposition of the urea into $NH_3$ occurs. The $NH_3$ reacts then over the SCR catalyst with NO and $NO_2$ to $N_2$ and water. Depending on the NO to $NO_2$ ratio, the SCR reaction occurs via the slow (1), normal (2) and fast (3) reaction pathway [3].

$$8NH_{3(g)} + 6NO_{2(g)} \rightarrow 7N_{2(g)} + 12H_2O_{(g)} \tag{1}$$

$$4NH_{3(g)} + 4NO_{(g)} + O_{2(g)} \rightarrow 4N_{2(g)} + 6H_2O_{(g)} \tag{2}$$

$$2NH_{3(g)} + NO_{(g)} + NO_{2(g)} \rightarrow N_{2(g)} + 3H_2O_{(g)} \tag{3}$$

The catalytic material of which the SCR catalyst consists are often zeolites, such as SSZ-13 [5–7] and ZSM-5 [8,9], containing transition metals like Fe or Cu. The exact structure of the catalytically active site in SCR is still heavily under debate, but likely the copper is present as a heterogeneous mixture of CuOx species and mono- and bi-atomic species coordinated to the Brønsted acid sites (BAS) [10–12] of the zeolite. The BAS of the zeolite act as adsorption sites of excess $NH_3$.

In order to reduce $NO_x$ over the catalyst, ammonia is required. The $NH_3$ is generated by decomposition of urea introduced as a 32 wt% aqueous solution. When urea is overdosed on the system, the aqueous urea solution is transformed in solidified products named urea deposits. Previously, we have reported on the effect of temperature and time on the formation of such solid material and the molecular composition of them [4] in the urea mixing part in front of the SCR catalyst. However, overdosing of the SCR urea injection fluid can also lead to urea adsorption at the SCR catalyst surface, potentially deactivating the catalyst. Therefore, it is essential to gain insights on the behavior of adsorbed urea at the catalyst surface at various temperatures in the SCR operation window. Furthermore, from the gained knowledge, it is possible to identify which urea derived molecules are being formed upon heat treatment and it provides insides to develop novel regeneration strategies.

In order to develop new regeneration strategies for exhaust aftertreatment systems, an initial simple and easy laboratory screening is desirable. In this way, expensive and time-consuming test cell experiments can be reduced, making calibration of these systems substantially easier. Besides improving SCR efficiency, large amounts of test cell time is spent on developing strategies for SCR catalyst regeneration during operation and overcoming unexpected events, such as urea overdosing. For this purpose, FTIR equipment is present in almost every laboratory enabling such method to a large audience. In this study, an easy and quick FTIR method is presented to investigate the effect of urea decomposition after urea being impregnated inside the Cu/HZSM-5 zeolite framework. In this way, soaking of a SCR catalyst with a urea diesel injection fluid is simulated. The results found in this paper, accompanied with the reported literature references clearly presented in tables, can be used as a guideline for setting up new methods to speed up SCR calibration work. Besides providing a method to help SCR calibration, a case study is presented for the removal of urea after soaking a SCR Cu/HZSM-5 catalyst substrate with urea diesel injection fluid. This is a potential real life event which could, for instance, occur upon dosing malfunction. To better understand which phase in Cu/HZSM-5 contributes to the formation or adsorption of certain species upon urea decomposition, HZSM-5 and copper oxide (CuOx) have been impregnated with urea and underwent the same treatment. The obtained spectra were analyzed using absorbance band references to various urea derived species potentially being adsorbed on the catalyst surface, as reported in the literature and are shown in Table 1. Careful analysis of FTIR spectra of Cu/HZSM-5 shows that urea is completely decomposed at temperatures as low as 133 °C, well below typical operation temperatures of an SCR catalyst. Decomposition of urea at and below 250 °C showed the presence of adsorbates at the catalyst surface which are likely to be decomposition products of urea such as $NO_x$ related adsorbates, HNCO and $NH_3$ present at the catalyst surface. Above 250 °C, these adsorbates could not be observed. Also, the renderability of the material towards urea adsorption has been investigated by exposing the materials to air consisting of 10% humidity under flow and heating to 400 °C. From the FTIR spectra before and after treatment, it was shown that it is possible to fully remove urea even when high amounts of urea were initially present. These results show that it is possible to fully remove urea adsorbed at a Cu/HZSM-5 catalyst within the SCR operation window in exhaust gas aftertreatment systems after being soaked with a urea diesel injection fluid.

**Table 1.** Literature references to various FTIR absorbance bands.

| Species | HZSM-5 cm$^{-1}$ | Cu/HZSM-5 cm$^{-1}$ | CuOx cm$^{-1}$ | Reference |
|---|---|---|---|---|
| Physisorbed water | 1630 | 1630 | 1630 | [13] |
| Framework vibrations | 1873, 1967, 1637 | 1873, 1967, 1637 | | [14] |
| NH$_4^+$ symmetric bending vibration | 1463 | | | [15] |
| C=O Hydrogen bonded urea | 1708 | | | [16] |
| Carbonate species | 1765 | | 1397 | [17] |
| Monodentate nitrite | | | 1423 | [18] |
| Monodentate nitrite | | | 1450 | [19] |
| Lewis acid sites | | | 1620 | [20] |
| C=N stretch | | | 1696 | [21] |
| Uretdion groups | | | 1780 | [22] |
| HNCO | | | 2330 | [23] |
| C=O stretching frequency | 1720 | | | [24] |
| Stretching of C=O and bending of NH$_2$ | 1680 | | | [25,26] |
| Tautomeric forms of urea C-N to C=N | 1650 | | | [27] |
| C=O correlated to amide | 1650 | | | [28] |
| Mono/bi dentate nitrate | | 1583 | | [29] |
| Chelated nitro species | | 1515, 1545, 1580 | | [30] |
| Asymmetric NO$_2$ | | 1480 | | [29] |
| N$_2$O$_3$ | | 1684 | | [29] |
| Mono dentate nitrate | | 1430–1447 | | [29] |
| vNO-Cu complex | | 1708 | | [31] |
| Cu$^+$-ZSM-5-(NO)$_2$ | | 1734 | | [31] |
| Cu$_2$O | | 1709 | | [31] |
| Cu$^+$-NO | | | 1763 | [32] |
| *cis*-HNO$_2$ | | | 1670 | [33] |

## 2. Experimental

### 2.1. Sample Preparation

Zeolite ZSM-5 with and Si/Al ratio of 15 in ammonia form was purchased from Alfa Aesar. To obtain the proton form, the zeolite was calcined by heating with a gradient of 5 °C/min to 450 °C and was kept at this temperature for 4 h [34]. A part of the proton form zeolite was impregnated with a CuSO$_4$ (Merck) solution via incipient wetness impregnation to introduce Cu species to the zeolite material. It was aimed to achieve 5 wt% of Cu in the final zeolite material. The copper oxide was prepared by calcining CuSO$_4$ in air by heating to 700 °C with a gradient of 5 °C/min and keeping at 700 °C for 6 h. The proton form of the ZSM-5 zeolite materials is denoted as HZSM-5, whereas the Cu containing zeolite ZSM-5 is denoted as Cu/HZSM-5. The calcined CuSO$_4$ material is coded as CuOx.

An aqueous 32 wt% urea solution was prepared by dissolving the appropriate amount of urea (Merck) in demineralized water and was introduced into the prepared zeolite and oxide materials using incipient wetness impregnation. Prior to impregnation, the materials were dried in a static oven overnight at 110 °C. The incipient wetness impregnation procedure was repeated until the desired zeolite or oxide to urea weight ratio was obtained.

*2.2. Analysis*

Powder X-ray diffraction measurements were collected under ambient temperature using a Malvern[®] Panalytical Empyrean at a voltage of 45.0 kV and a current of 40 mA for Cu Kα (l = 1.5418 Å), with a step size of 0.0131° in 2θ and a range of 3–60° in 2θ and a total scan time of 62 min.

The $N_2$ physisorption experiments were measured using a Micromeritics ASAP2020 instrument at −196 °C. Prior to analysis, samples were degassed at 120 °C for 8 h. The surface areas were calculated from the adsorption branch in the 0.001–0.2 $p/p_0$ region via the Brunauer-Emmett-Teller (BET) method [35]. The t-plot method was used to calculate micropore volumes and pore size distribution [36].

The FTIR spectra were measured on a Perkin Elmer Spectrum II machine. Prior to FTIR analysis, the samples were dried overnight at 100 °C to minimize water interfering in the spectra but preventing urea from being decomposed. After drying, the samples were pressed into pellets using a die. The applied pressure to the pellets was 5 tonnes and after pelletizing the materials were directly transferred to the FTIR machine and measured in transmission mode at room temperature. The obtained spectra were analyzed using Spectrum II software (version 2017). To deconvolute the obtained spectra Fytik fitting (version 1.3.1) software was used. The spectra were fitted using Gaussian peaks.

To conduct the regeneration experiments under humid conditions, 100 mg of powdered material was loaded in a stainless-steel tube reactor and placed in a custom-made oven. The powdered material was first impregnated with urea, followed by heat treatment at 200 °C for 4 h. The amount of urea being impregnated to the Cu/HZSM-5 material was varied from 1:0.5 wt/wt ratio to 1:3. After the reactor being loaded with the material, an air gas flow of 30 mL/min was fed to the reactor. To increase humidity, the air stream was led through a water filled saturator, kept in a heating bath to obtain the desired humidity (10%). During the experiment, the reactor temperature was increased (5 K/min) to 673 K and kept for 1 h. After the experiment, the reactor was left to cool to room temperature. The remaining material was collected, pelletized and measured in FTIR.

## 3. Results

*3.1. Characterization*

The crystallographic properties of the prepared HZSM-5 and Cu/HZSM-5 materials before and after impregnation are shown in Figure 1. The diffraction patterns of all zeolites reveal reflections which strongly correlate to that of HZSM-5 [37,38]. Also, the absence of a broad reflection between 15° and 30° 2θ indicates the absence of an amorphous phase [39]. This indicates that, after incipient wetness impregnation of the HZSM-5 sample with Cu solution followed by calcination, the crystal structure was preserved. Furthermore, some small extra reflections are observable at 35.5 and 42.4 from the Cu/HZSM-5 sample, indicating $CuO_x$ particles [40].

The textural properties of the prepared materials were analyzed by conducting $N_2$ physisorption experiments. From the adsorption branch, the surface area and pore volume values have been calculated and are presented in Table 2. The reference ZSM-5 material contained a micropore volume of 0.11 $cm^3/g$ which is similar to values reported in literature [41]. After impregnation with urea, the material slightly reduced in micro pore volume and surface area being, 0.09 $cm^3/g$ and 170 $m^2/g$. After introduction of Cu into the ZSM-5 zeolite, the materials micro pore volume has reduced to 187 $m^2/g/$. Introduction of urea in the Cu-ZSM-5 material resulted in less reduction in $S_{mic}$ (7%) compared to the Cu deficient material (18%). The $CuO_x$ material reveals to possess a very low surface and no micro pores have been identified by the physisorption experiments. The low micropore volume has been reported for metal oxides, such as $CuO_x$ [42].

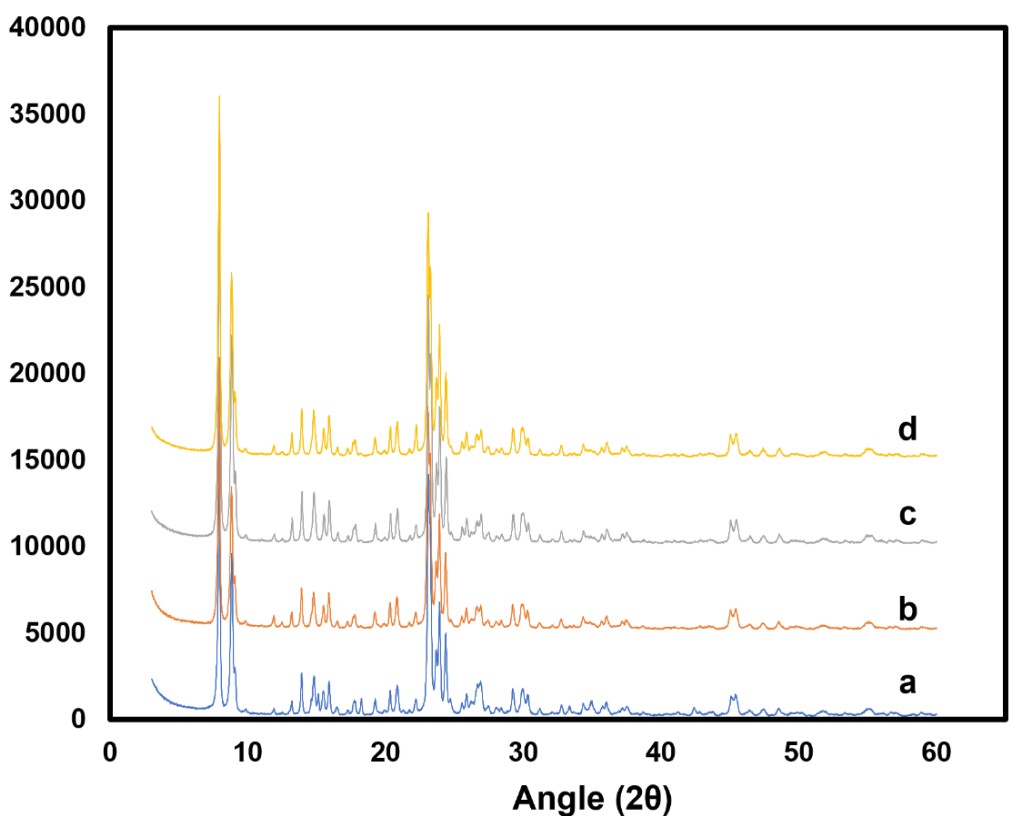

**Figure 1.** X-Ray diffraction patterns of (a) Cu/HZSM-5, (b) Cu/HZSM-5-urea impregnated, (c) HZSM-5 and (d) HZSM-5-urea impregnated.

**Table 2.** Physico-chemical properties of the CuOx, ZSM-5 and Cu/HZSM-5 materials.

| Sample | | $S_{tot}$ (m²/g) | $S_{BET}$ (m²/g) | $S_{mic}$ (m²/g) | $V_{mic}$ (cm³/g) | $V_{tot}$ (cm³/g) |
|---|---|---|---|---|---|---|
| ZSM-5 | - | $342 \pm 9$ | $329 \pm 8$ | $207 \pm 5$ | $0.11 \pm 0.01$ | $0.27 \pm 0.01$ |
| | Urea | $265 \pm 7$ | $254 \pm 6$ | $170 \pm 4$ | $0.09 \pm 0.01$ | $0.21 \pm 0.01$ |
| Cu/HZSM-5 | - | $293 \pm 7$ | $281 \pm 7$ | $187 \pm 5$ | $0.10 \pm 0.01$ | $0.24 \pm 0.01$ |
| | Urea | $276 \pm 7$ | $265 \pm 7$ | $174 \pm 4$ | $0.10 \pm 0.01$ | $0.22 \pm 0.01$ |
| CuOx | - | $9 \pm 1$ | $8 \pm 0$ | 0 | 0 | $0.06 \pm 0.01$ |
| | Urea | $7 \pm 1$ | $6 \pm 0$ | 0 | 0 | $0.09 \pm 0.01$ |

*3.2. FTIR Analysis after Heat Treatment*

The infrared spectra measured for the heat-treated urea impregnated CuOx materials are presented in Figure 2A. The CuOx sample treated at 133 °C contains an absorbance peak at 1415 cm$^{-1}$ and a shoulder band at 1430 cm$^{-1}$. These bands have been related to NO dentate species coordinated to metal oxide species [29]. The bands present at 1515 cm$^{-1}$ and 1573 cm$^{-1}$ can be attributed to chelated nitro species [30]. A small band at 1628 cm$^{-1}$ can be attributed to bidentate nitrate species [30], while the feature at 1670 cm$^{-1}$ was suggested to refer to *cis*-HNO$_2$ [33]. The band at 1709 cm$^{-1}$ has been attributed to NO adsorbed to Cu$_2$O [9] and the shoulder band at 1763 cm$^{-1}$ was attributed to Cu$^+$-NO species [32]. A small absorption feature is observed at 2349 cm$^{-1}$ which is attributable to chemisorbed CO$_2$ [43]. The described features are more intense after heating the urea impregnated CuOx material at 150 °C suggesting more of such species to be present at the catalyst surface. Interestingly, after heating at 200 °C most features are somewhat lower in intensity compared to that of the spectrum recorded for the sample treated at 150 °C. The bands at 1415 cm$^{-1}$ and 1430 cm$^{-1}$ become substantially more intensive, suggesting a lower concentration of metal coordinated NO dentate species. The sample treated at 250 °C did only show a large feature in the 1415–1430 cm$^{-1}$ region. The samples treated

at 300 °C and 350 °C did not show any features. Interestingly, the spectrum prepared at 400 °C contains negative at 1690, 1454, 1414 and 1400 cm$^{-1}$.

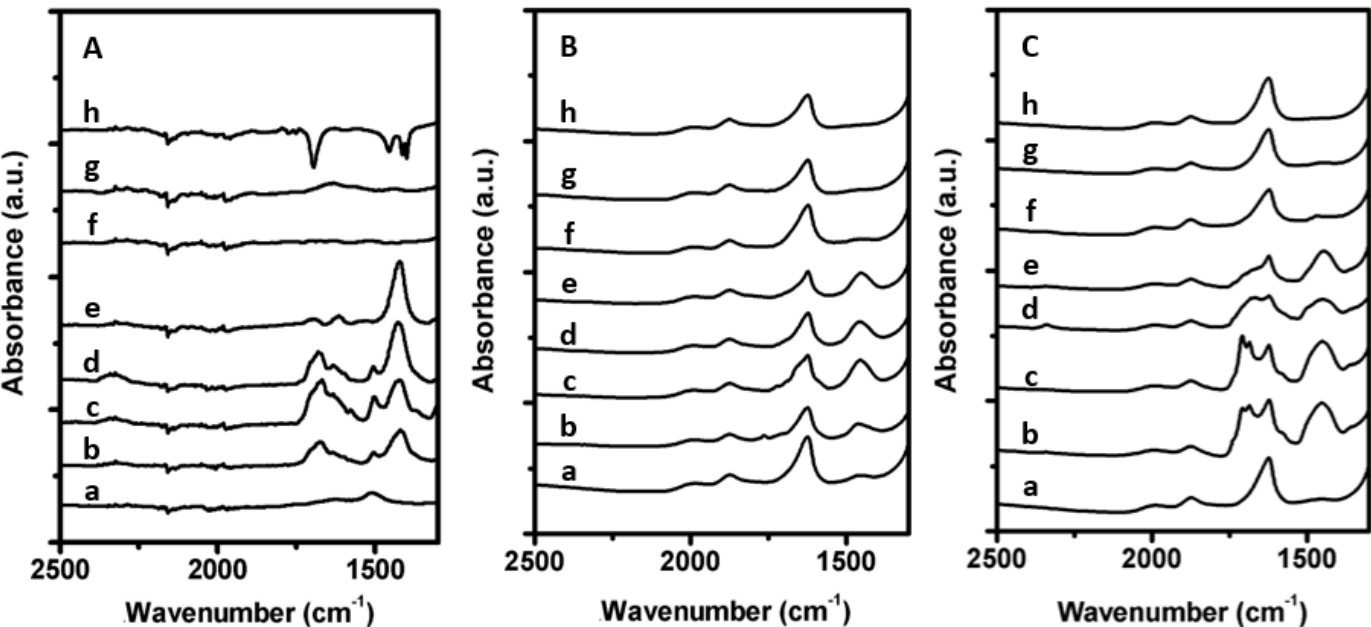

**Figure 2.** Infrared spectra of (**A**) CuOx, (**B**) HZSM-5 and (**C**) Cu/HZSM-5 representing of the (a) parent materials and the materials being impregnated with urea and heat treated for 4 h at (b) after 133 °C, (c) 150 °C, (d) 200 °C, (e) 250 °C, (f) 300 °C, (g) 350 °C and (h) 400 °C.

Figure 2B shows the infrared spectra of HZSM-5 zeolites impregnated with urea and heated at temperatures varying from 133–400 °C. The initial zeolite to urea weight ratio (Z/U) of the impregnated HZSM-5 zeolite was 1. Evaluation of the HZSM-5 related spectra in Figure 2B reveals a strong absorbance band at 1630 cm$^{-1}$ which can be related to adsorbed water [13]. Also, vibrations relating to zeolitic framework vibrations [14] at 1873 cm$^{-1}$ and 1967 cm$^{-1}$ can be observed in the spectra. The absorbance spectrum recorded for the parent HZSM-5 material contains a small band at 1463 cm$^{-1}$ previously related to $NH_4^+$ chemisorbed to the BAS [15]. The presence of such species could be explained by the presence of residual $NH_4^+$ left after the calcination process of NH$_4$-ZSM-5 to prepare the proton form (HZSM-5). After impregnation and heating at 133 °C, an increase is observable in intensity of the 1463 cm$^{-1}$ band, suggesting a higher concentration of $NH_4^+$ absorbed tot the BAS. A new small feature is observable at 1708 cm$^{-1}$ which can be related to the C=O stretch vibration of hydrogen bonded urea [15]. Also, a small absorbance band at 1765 cm$^{-1}$ is present in the FTIR spectrum, which corresponds to C=O stretch vibration. After treatment, a 150 °C the band at 1463 cm$^{-1}$ representative for $NH_4^+$ coordinated to BAS is substantially larger. New small shoulder bands are observable in the FTIR spectra at 1650 cm$^{-1}$, 1680 cm$^{-1}$ and 1720 cm$^{-1}$. The absorbance shoulder band of 1650 cm$^{-1}$ is close to that of compounds containing C=N bond bonds and suggests the presence of a tautomeric form of urea, which was also earlier reported by Piasek et al. [27]. Also, people have attributed this absorbance band to the presence of C=O groups in the amide bond. The 1680 cm$^{-1}$ band was attributed to both C=O stretching and NH$_2$ bending modes [25,26]. Heat treatment of the urea impregnated H-ZSM-5 at 200 °C reveals less absorbance bands in the FTIR spectrum compared to that of samples prepared at 133 °C and 150 °C. The absorbance band at 1463 cm$^{-1}$, which has been attributed to $NH_4^+$ species adsorbed at the BAS [15]. The intensity of this band is similar to that of urea impregnated H-ZSM-5 treated at 150 °C. Increasing the treatment temperature further to 250 °C reveals a decrease in the intensity of the 1463 cm$^{-1}$ peak and is strongly decreased for the samples prepared at 300 °C and higher.

The intensities of the absorbance band at 1463 cm$^{-1}$ were determined by deconvolution of the absorbance band using fitting software. The values obtained upon deconvolution were normalized to that of the spectrum with the highest intensity, in this case that for the material prepared by heat treatment at 150 °C after impregnation with urea solution (Table 3). The results reveal a slow decrease in intensity up to a treatment temperature of 250 °C. Only a small feature at 1463 cm$^{-1}$ is observable for the samples treated at 300 °C and 350 °C and is not observable for the sample treated at 400 °C.

**Table 3.** Deconvolution results of the spectra recorded for the various species. The deconvolution was repeated 3 times and results were in the 3% error margin of the obtained areas.

| Sample | T (K) | 1430–1530 cm$^{-1}$ | 1430–1447 cm$^{-1}$ | 1466 cm$^{-1}$ | 1480 cm$^{-1}$ | 1515 cm$^{-1}$ |
|---|---|---|---|---|---|---|
| HZSM-5 | - | 0 | 0 | 0 | 0 | 0 |
| | 133 | 29 | 0 | 29 | 0 | 0 |
| | 150 | 100 | 0 | 100 | 0 | 0 |
| | 200 | 76 | 0 | 76 | 0 | 0 |
| | 250 | 68 | 0 | 68 | 0 | 0 |
| | 300 | 8 | 0 | 8 | 0 | 0 |
| | 350 | 4 | 0 | 4 | 0 | 0 |
| | 400 | 0 | 0 | 0 | 0 | 0 |
| Cu/HZSM-5 | - | 0 | 0 | 0 | 0 | 0 |
| | 133 | 99 | 40 | 44 | 12 | 3.9 |
| | 150 | 100 | 46 | 35 | 19 | 0.5 |
| | 200 | 79 | 48 | 39 | 8 | 3 |
| | 250 | 64 | 52 | 36 | 7 | 3 |
| | 300 | 7 | 21 | 78 | 0 | 0 |
| | 350 | 0 | 0 | 0 | 0 | 0 |
| | 400 | 0 | 0 | 0 | 0 | 0 |

Figure 2C shows the spectra of urea impregnated Cu/HZSM-5 materials recorded by FTIR. Investigating the spectra of the Cu/HZSM-5 sample series reveals that spectral line Cu/ZSM-5 represents a similar line shape to that of HZSM-5. After impregnation of Cu/HZSM-5 with urea and heat treatment at 133 °C, new absorbance bands can be observed. A broad absorbance feature is observed in the 1400–1530 cm$^{-1}$ region with the maximum centered at 1448 cm$^{-1}$. The broad band consists of various smaller bands. Upon deconvolution bands can be fitted in the 1430–1447 cm$^{-1}$ region representative of monodentate nitrate species formed upon absorbance to Cu-O [29]. Further convolution bands are located at 1463 cm$^{-1}$ and 1480 cm$^{-1}$ representative of NH$_4^+$ coordinated to the BAS [15] and NO$_2$ species chelated to Cu species [30], respectively. Also, strong sharp absorbance bands can be observed at 1684 cm$^{-1}$ representative for N$_2$O$_3$ species [29] and 1709 cm$^{-1}$ attributable to NO adsorbed to Cu$_2$O species [31]. A shoulder band can be observed at 1734 cm$^{-1}$ which can be correlated to two NO species adsorbed to Cu$^+$ in ZSM-5 [31]. The features become more pronounced when treating the Cu/HZSM-5 material at 150 °C. The features become less substantial when increasing the treatment temperature further to 200 °C and 250 °C. A spectral feature for the 200 °C treated sample can be observed at 2349 cm$^{-1}$ which has been attributed to chemisorbed CO$_2$ species related to copper oxide species [43]. The spectra recorded for the Cu/HZSM-5 materials treated at 300 °C and higher only contain features characteristic for zeolite framework vibrations.

The intensities of the absorbance band between 1430 and 1515 cm$^{-1}$ have also been deconvoluted for the various NO$_x$ and NH$_3$ species present at the catalyst surface. The intensities of the urea impregnated Cu/HZSM-5 materials treated at various temperatures have been compared to each other. Similar to the HZSM-5 sample series, the largest intensity was obtained for the Cu/HZSM-5 urea impregnated samples heat treated at 150 °C. Therefore, the intensities of the broad 1430–1515 cm$^{-1}$ band of the other samples have been related to that of urea impregnated Cu/HZSM-5 treated at 150 °C. The broad band was deconvoluted by fitting peaks at 1515 cm$^{-1}$ (chelated nitro species) [30], 1480 cm$^{-1}$ (asymmetric NO$_2$) [29], 1466 cm$^{-1}$ (NH$_4^+$-BAS) [15] and 1430–1447 cm$^{-1}$ (monodentate nitrate) [29]. The results are depicted in Table 3. Interestingly, the intensity of

the 1430–1530 cm$^{-1}$ band for the Cu/HZSM-5 treated at 133 °C is similar to that of the sample treated at 150 °C. When comparing the percentages of the various species present by deconvolution of the broad 1430–1530 cm$^{-1}$ band reveals a higher concentration of NH$_3$ to be present adsorbed to the BAS for the Cu/HZSM-5 sample treated at 133 °C compared to the sample prepared at 150 °C. The concentrations of the various NO$_2$ species are seemingly suggesting oxidation of NH$_3$ to NO$_2$ at higher temperatures, which is likely catalyzed by the presence of Cu species [44]. Literature reports the CuOx species are likely to be responsible for the oxidation of NH$_3$ to NO$_2$ [44,45]. Further increase in the treatment temperature to 200 °C leads to a reduction in the 1430–1530 cm$^{-1}$ band intensity.

### 3.3. Effect of Heating Time

The Cu/HZSM-5 material prepared at 200 °C showed the presence of various species at the catalyst surface when treated for 4 h. To investigate whether the adsorbed species are removed over time, the urea impregnated Cu/HZSM-5, CuOx and HZSM-5 measured in FTIR after 2, 4, 6 and 24 h. The FTIR spectrum recorded for CuOx after 2 h (Figure 3A) reveals the presence of various bands at 1397 cm$^{-1}$, 1450 cm$^{-1}$, 1620 cm$^{-1}$, 1696 cm$^{-1}$ and 1780 cm$^{-1}$. The band at 1397 cm$^{-1}$ corresponds to the formation of CO$_3^{2-}$ species [17], while the 1456 cm$^{-1}$ has been related to nitrate species [19]. The band at 1620 cm$^{-1}$ can be related to Cu lewis acid sites which formed complexes with NH$_3$ [20] and the 1696 cm$^{-1}$ is attributable to C=N stretch vibrations [21]. The 1780 cm$^{-1}$ band corresponds to uretdione groups being an isocyanate dimer [22]. After heating for 4 h, bands corresponding more to that of NO adsorbed species interacting with CuOx are visible. Also a band representable for HNCO is observable at 2330 cm$^{-1}$ [23]. After 6 h of heating, a decrease in the intensity of adsorbed NO-species bands can be observed, suggesting less adsorbed molecules to the surface. After 24 h almost all spectral features have disappeared, except for a band at 1423 cm$^{-1}$ corresponding to some adsorbed NO-species [18].

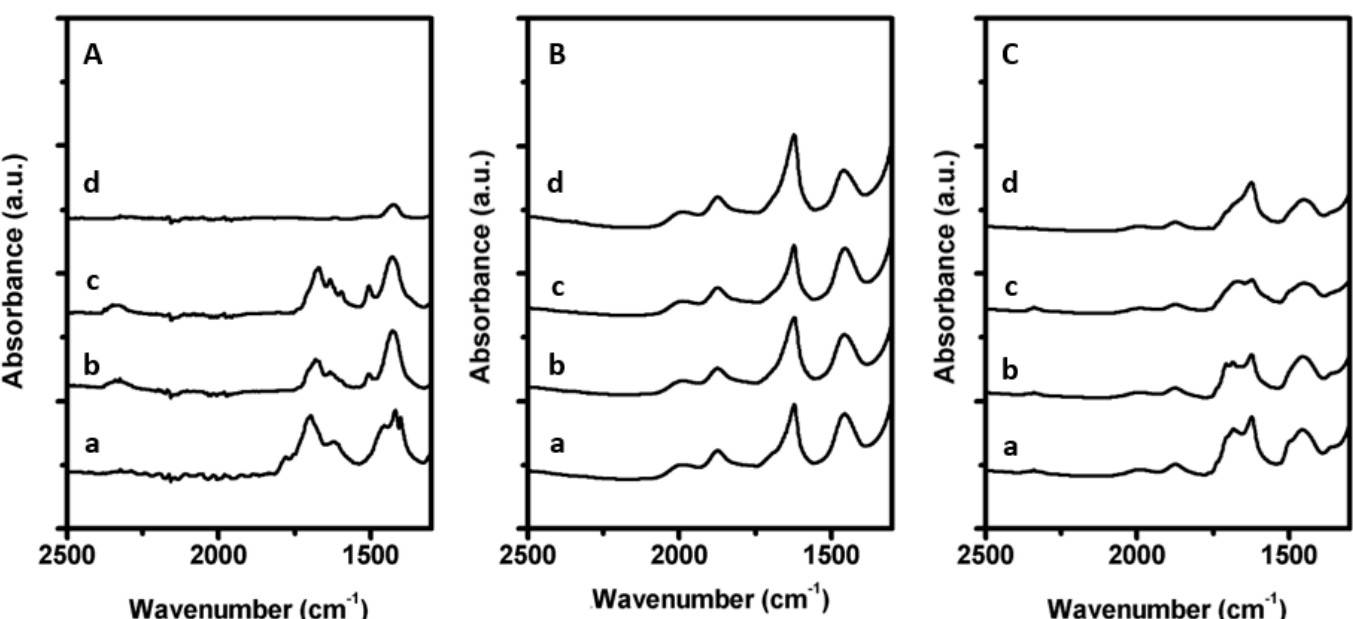

**Figure 3.** Infrared spectra of (**A**) CuOx, (**B**) HZSM-5 and (**C**) Cu/HZSM-5 after impregnation with a 32 wt% aqueous urea solutions and heated at 200 °C for various time being (a) 2 h, (b) 4 h, (c) 6 h and (d) 24 h.

Urea impregnated HZSM-5 materials were also measured using FTIR, after exposure to the same time intervals (Figure 3B). The HZSM-5 sample treated for 2 h did not contain spectral features representing to that of urea. Only bands characteristic for zeolite ZSM-5 and of NH$_3$ adsorbed to the BAS (1463 cm$^{-1}$) were observable [15]. Spectra of HZMS-5 recorded

after exposure up to 6 h did not reveal lower intensities of the absorbance bands. After 24 h the intensity of the ammonia band is slightly decreased, suggesting some desorbance of $NH_3$ over time from the catalyst surface. The ammonia adsorbed to the ZSM-5 zeolite remains strongly adsorbed to the BAS at such temperatures. This corresponds to reported TPD results presenting $NH_3$ desorption to start just above 200 °C [46].

The FTIR spectra of Cu/HZSM-5 (Figure 3C) reveals a mixture of species adsorbed at the catalyst surface. Evaluation of the spectra collected for samples prepared at 200 °C over time, reveal that the adsorbates correlated to NO-Cu species can be desorbed even at relatively low SCR operation temperatures. Analysis of the spectra recorded for the Cu/HZSM-5 catalyst reveals the presence of adsorbed NO species and ammonium species as well adsorbed HNCO. The intensities of the spectral features indicative for the adsorbed $NO_x$ species become lower in intensity with longer treatment time and are weakly visible in the spectrum after 24 h of treatment. The band indicative for $NH_3$ remains substantially after 24 h of heat treatment.

*3.4. Regeneration under Realistic Conditions*

To investigate the ability of the prepared materials to desorb components originating from urea decomposition, the catalysts were exposed to gradual heating from room temperature to 400 °C and kept at this temperature for 1 h. To simulate humidity of an exhaust gas flow, the air flow contained 10% water. Also, the materials were loaded with various amounts of urea to investigate the effect of the amount of urea present in the zeolite materials. The spectra are shown in Figure 4. The spectra recorded for Cu/HZSM-5 shows bands of $NO_x$-species adsorbed to the catalyst surface to become more intense with increasing amount of urea loaded to the catalyst after 4 h of heating at 200 °C. After treatment under exhaust-like conditions, catalysts only showed features which resemble vibrations characteristic for Cu/HZSM-5, HZSM-5 and CuOx. From these results it is shown that the surface of the Cu/HZSM-5 catalyst can be cleaned from adsorbed molecules originating from decomposed urea deposits.

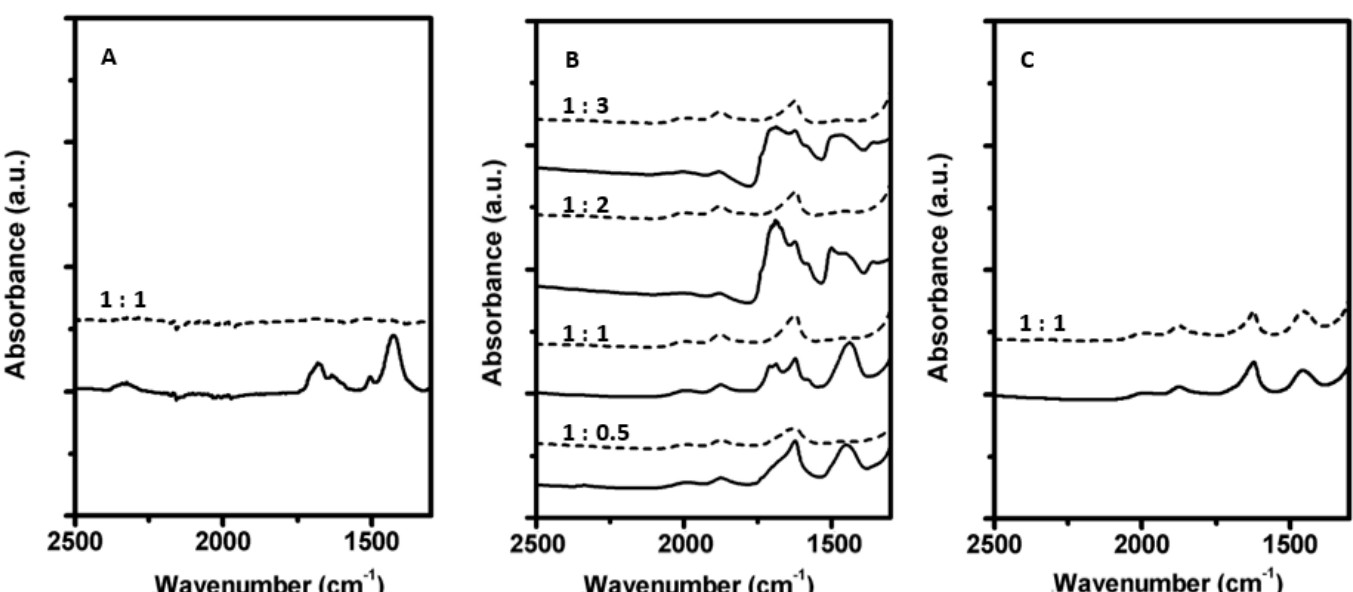

**Figure 4.** Infrared spectra of (**A**) CuOx, (**B**) Cu/HZSM-5 and (**C**) HZSM-5 before and after treatment in gradient heating from room temperature to 400 °C under an air flow consisting of a humidity of 10%. The weight ratio of Cu/HZSM-5: urea has been varied as indicated by the ratios depicted above the spectra.

## 4. Discussion

### 4.1. Temperature Effect on Urea Decomposition

The FTIR analysis performed on the HZSM-5, CuOx and Cu/HZSM-5 phase revealed that heat treatment of the urea impregnated materials led to quick decomposition of the urea at temperatures as low as 133 °C. The urea is likely to be readily decomposed into ammonia and HNCO according to mechanisms as reported in various literature. This is also evidenced by the presence of these species in the FTIR spectra recorded for Cu/HZSM-5 in the 133–250 °C temperature region. These species are each adsorbed to a different phase present in the Cu/HZSM-5 material. The adsorption of ammonia occurs due to interaction with the BAS present in the HZSM-5 phase, while the absorbance of HNCO can be assigned to the CuOx phase. Also, the presence of $NO_x$ and metal nitrate species were observed in the spectra of Cu/HZSM-5 as well in the spectra of CuOx but absent in the spectra recorded for HZSM-5. Therefore, it is likely to assume that these species are adsorbed to the CuOx phase present in Cu/HZSM-5. The formation of such $NO_x$ species and $NO_3^-$ species occurs likely due to oxidation of the formed $NH_3$ upon decomposition of urea over the present CuOx species. Quantification of the $NH_3$ band at 1463 cm$^{-1}$ via deconvolution reveals the maximum $NH_3$ absorbance capacity of the HZSM-5 to be reached at urea decomposition temperatures of 150 °C. In the temperature window of 150–250 °C the concentration of $NH_3$ adsorbed to the zeolite decreases slightly with increasing temperature and above 250 °C the concentration of adsorbed $NH_3$ decreases rapidly. A similar observation was made for Cu/HZSM-5 for the ammonia adsorption. This trend coincides with observations from Temperature Programmed Desorption experiments reported earlier in literature [13,14] using ammonia as probe molecule. The intensity of the band indicative for the adsorbed $NO_x$ species in the spectra of Cu/HZSM-5 follows a similar trend with treatment temperature showing no spectral features which could be related to such species above 250 °C. This behavior could be explained by the instability of the various $NO_x$ type of Cu species at elevated temperatures. For instance, the decomposition temperature for $CuNO_3.2H_2O$ is reported to be approximately 160–180 °C [47–49]. Another example is the decomposition of $Cu(NO_3)_2$ taking place in the 200–280 °C temperature window [50].

### 4.2. Regenerability of Cu/HZSM-5

Knowledge on the regeneration properties of SCR catalysts is essential to understand their suitability for real world applications. The removal of urea deposited on the catalyst substrate is essential information to predict catalyst lifetime. Studying the decomposition and desorption/adsorption of various molecular species over time at 200 °C, reveals that the Cu/HZSM-5 desorbs adsorbed $NO_x$-species, HNCO and $NH_3$ species at a very low rate. The low desorption rate of $NH_3$ correlates well to the observations made for $NH_3$ desorption from HZSM-5 and can be explained by the fact that such events take place at 250 °C and higher [51,52]. When looking at the desorption of adsorbed $NO_x$-species it is observable that desorption from the CuOx materials occurs at a higher rate. This suggests that these species are more weakly bound to the pure CuOx phase compared with the Cu/HZSM-5 phase. Possibly, the confinement structure of the zeolite contributes to this discrepancy in adsorption strength. Testing the desorption of the catalysts at high temperature (400 °C) and humid conditions in flow experiments to simulate regeneration under exhaust gas flow revealed that it is possible to completely remove adsorbed molecules related to the decomposition of urea from the Cu/HZSM-5 catalyst surface as well from the CuOx and HZSM-5 surfaces.

## 5. Conclusions

In this study, a straightforward FTIR method was presented to investigate the heat treatment effect of Cu/HZSM-5 materials after introduction of urea solid deposits prepared via incipient wetness impregnation. This, to simulate the unlikely event of urea soaking when applied as SCR catalyst in aftertreatment equipment for the reduction of $NO_x$ present in the exhaust gas of heavy-duty equipment run on diesel. It was shown that the full

decomposition of urea, upon heat treatment, already occurs at relatively low temperatures (133 °C). At temperatures below 250 °C, $NO_x$ related species, $NH_3$ and HNCO were observed in the FTIR spectra of Cu/HZSM-5, suggesting them to be adsorbed. The FTIR spectra show the presence of $NH_3$ species adsorbed at the BAS of the HZSM-5 phase and this is likely to be a product of urea decomposition. The spectra also show the presence of chelated NO and $NO_2$ as well as Cu-nitrates as a result of interaction with Cu species. These $NO_x$ species and copper nitrates are possibly formed due to oxidation of $NH_3$ formed during thermolysis of urea. The concentration of these adsorbed species decreases at increased thermolysis temperature and are no longer visible at temperatures >250 °C. The rate of decomposition was seen to be strongly dependent on the treatment temperature. Extended heat treatment at 200 °C revealed only minor loss of absorbents after 6 h and were still observable in the FTIR spectra after 24 h. Simulating catalyst regeneration from urea adsorption under SCR operation conditions was done under continuous air flow with a humidity of 10% and at elevated temperatures (400 °C). These findings showed the complete removal of adsorbents from the Cu/HZSM-5 surface even at substantial loadings of urea to the catalyst. The results presented in here show the possibility of removing the urea from the catalyst after being introduced into zeolite structure without leaving adsorbents on the catalyst surface.

**Author Contributions:** Conceptualization, C.T. and V.D.; methodology, C.T., B.e.A., J.S., R.S., V.D.; formal analysis, C.T., B.e.A., J.S., R.S., V.D.; investigation, C.T., B.e.A., J.S., R.S., V.D.; resources, C.T.; data curation, C.T., B.e.A., J.S., R.S., V.D.; writing—original draft preparation, C.T., B.e.A., J.S., R.S., V.D.; writing—review and editing, C.T., V.D; supervision, C.T.; project administration, C.T. All authors have read and agreed to the published version of the manuscript.

**Funding:** This research received no external funding.

**Data Availability Statement:** The data presented in this study are available this article.

**Conflicts of Interest:** The authors declare no conflict of interest.

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
