# Peer review of "Decomposition of Heavy Diesel SCR Urea Fluid Adsorbed in Cu/HZSM-5 SCR Catalysts Studied by FTIR Spectroscopy at Ambient Conditions"

_reactions, doi:10.3390/reactions3040038_

Round 1

Reviewer 1 Report

Reviewer comment: Thanks for inviting me to review this paper titled " Decomposition of heavy diesel SCR urea fluid adsorbed in Cu/HZSM-5 SCR catalysts studied by FTIR spectroscopy at ambient conditions". In this paper, the authors study the decomposition of heavy diesel using Selective Catalytic Reduction (SCR). Authors used FTIR as a tool for chemical identification. This work has no novelty, with poor explanations and figures. Please be more careful about preparing a manuscript. Even though this manuscript is poorly written, it still can be improved for the possible reader of "Reactions". Therefore, I recommend publishing this research paper in the "Reactions", but only after a manuscript revision. Please find my comments below.

1.   The introduction should mention why this work and the methods are important.

2.   All the chemical concentrations must be mentioned clearly in FTIR analysis.

3.   No characterizations have been done, even though this paper is about SCR. XRD of fresh catalyst along with bare ZSM-5 should be added.

4.   FESEM of the prepared catalyst should be added with a proper explanation before and after the catalytic reactions.

5.   Why is the calcination temperature so high (700 ⁰C)? This could be done at 550 ⁰C to get metal oxide.

6.   It seems all the spectrums of IR have been taken as absorbance. Why was the Fig. 1(A)-h spectrum taken as transmittance?

7.   All the figures must be redrawn. Why 0.5 (Fig. 3(B)) is out of the figure and overlapped with the wavenumber? All the levelling of the IR spectrums must be clear with no overlapping.

      ……………….Good luck…………………

Author Response

Reviewer #1:

  1. The introduction should mention why this work and the methods are important.

The below text has been added to the manuscript to highlight the importance of this work. In particular, the importance of this work lies on the basis of an easy FTIR method. By this method, not only we showed the removal of the adsorbed urea from catalyst surface, but also developed a new regeneration strategy for automotive SCR catalysts. We presented a simple and straightforward method allowing initial studies at the laboratory level thereby reducing expensive and time consuming test cell experiments with life size engines and after treatment systems. In the revised manuscript this was more emphasized.

“In order to develop new regeneration strategies for exhaust aftertreatment systems an initial simple and easy laboratory screening is desirable. In this way expensive and time consuming test cell experiments can be reduced making calibration of these systems substantially easier. Besides improving SCR efficiency large amounts of test cell time is spend on developing strategies for SCR catalyst regeneration during operation and overcoming unexpected events such as urea overdosing. For this purpose FTIR equipment is powerful equipment present in almost every laboratory enabling such method to a large audience. In this study an easy and quick FTIR method is presented to investigate the effect of urea decomposition after urea being impregnated inside the Cu/HZSM-5 zeolite framework. In this way soaking of a SCR catalyst with an urea diesel injection fluid is simulated. The results found in this paper accompanied with the reported literature references clearly presented in tables can be used as guideline for setting up new methods to speed up SCR calibration work. Beside providing a method to help SCR calibration a case study is presented for the removal of urea after soaking a SCR Cu/HZSM-5 catalyst substrate with urea diesel injection fluid. This is a potential real live event which could for instance occur upon dosing malfunction. To better understand which phase in Cu/HZSM-5 contributes to the formation or adsorption of certain species upon urea decomposition, HZSM-5 and copper oxide (CuOx) have been impregnated with urea a follow the same treamtent. The obtained spectra were analyzed using absorbance band references to various urea derived species potentially being adsorbed on the catalyst surface as reported in literature and are shown in Table 1.  Careful analysis of FTIR spectra of Cu/HZSM-5 shows that urea is completely decomposed at temperatures as low as 133°C, well below typical operation temperatures of an SCR catalyst. Decomposition of urea at and below 250°C showed the presence of adsorbates present at the catalyst surface which are likely to be decomposition products of urea such as NOx related adsorbates, HNCO and NH3 present at the catalyst surface. Above 250°C these adsorbates could not be observed. Also the regenerability of the material towards urea adsorption has been investigated by exposing the materials to air consisting of 10% humidity under flow and heating to 400°C. From the FTIR spectra before and after treatment it was shown that it is possible to fully remove urea even when high amounts of urea were initially present. These results show that it is possible to fully remove urea adsorbed at a Cu/HZSM-5 catalyst after being soaked with an urea diesel injection fluid.”

  1. All the chemical concentrations must be mentioned clearly in FTIR analysis.

Chemical concentrations are given.

  1. No characterizations have been done, even though this paper is about SCR. XRD of fresh catalyst along with bare ZSM-5 should be added.

The focus in this paper was on the identification of urea species. N2 physisorption was performed to show the accessibility of the zeolite pores to the urea solution. To the revision XRD patterns of the Cu/HZSM-5 and HZSM-5 has been added before and after impregnation with urea (Figure 1). These patterns show complete preservation of the zeolite crystal structure.

  1. FESEM of the prepared catalyst should be added with a proper explanation before and after the catalytic reactions.

In this paper no SCR has been conducted – the aim of the paper was to understand if the urea could be removed from the catalyst surface. The measurements in FESEM would not make any contribution

to the state of the catalyst after removal of the urea.

  1. Why is the calcination temperature so high (700 ⁰C)? This could be done at 550 ⁰C to get metal oxide.

We used the precursor CuSO4 to simulate SCR catalysts exposed to real world conditions. In practical mobile aftertreatment applications SCR materials are regenerated at temperatures as high as 700°C. This is emphasized in the manuscript.

  1. It seems all the spectrums of IR have been taken as absorbance. Why was the Fig. 1(A)-h spectrum taken as transmittance?

It is a choice of presentation style.

  1. All the figures must be redrawn. Why 0.5 (Fig. 3(B)) is out of the figure and overlapped with the wavenumber? All the levelling of the IR spectrums must be clear with no overlapping.

The figures have been redrawn and improved in the updated manuscript.

Reviewer 2 Report

This paper presents a valuable FTIR method to study the decomposition of urea deposited on Cu/HZSM-5 SCR 10 catalysts and therewith the ability of the Cu/HZSM-5 SCR catalyst to be regenerated after possible excess doses of  SCR urea fluids during operation. In my opinion the reported results are not particularly surprising. Nonetheless, this research is interesting and I recommend their publication in "Reactions".

There are a couple of relevant points that should be improved prior to publication.

1) Abstract: more details about the current results should be given in the second part of the abstract. For example, the temperatures at which urea was observed to decompose on the different substrates should be specified already here. Moreover, it should be clearly explained here that the FTIR spectra were recorded on samples previously heated to high temperature and successively cooled again to room temperature, i.e. that spectra were not directly recorded on hot samples.

2) More literature references should be added in the experimental part. For example, I would recommend to add one or more references about the proton-form zeolite, about the Brunauer-Emmett-Teller (BET) method and about the t-plot method.

3) The quality of figures should be improved. Writings are often overlapped with spectra and the whole graphs are of a bad quality. I would moreover suggest to introduce a couple of images of the analyzed samples before presenting the spectra.

4) Uncertainty analysis is missing. The Authors should please present the error bands characterizing the reported concentrations, compositions, spectral features, and the FTIR instrumental uncertainty.

5) English can be improved, in terms of both grammar and style. 

Author Response

# Reviewer 2

1) Abstract: more details about the current results should be given in the second part of the abstract. For example, the temperatures at which urea was observed to decompose on the different substrates should be specified already here. Moreover, it should be clearly explained here that the FTIR spectra were recorded on samples previously heated to high temperature and successively cooled again to room temperature, i.e. that spectra were not directly recorded on hot samples.

The abstract has been adjusted according to the remarks made by the reviewer.

2) More literature references should be added in the experimental part. For example, I would recommend to add one or more references about the proton-form zeolite, about the Brunauer-Emmett-Teller (BET) method and about the t-plot method.

More references have been added to the experimental section.

3) The quality of figures should be improved. Writings are often overlapped with spectra and the whole graphs are of a bad quality. I would moreover suggest to introduce a couple of images of the analyzed samples before presenting the spectra.

The quality of the figures has been improved in the updated manuscript.

4) Uncertainty analysis is missing. The Authors should please present the error bands characterizing the reported concentrations, compositions, spectral features, and the FTIR instrumental uncertainty.

A statement on the error margin for the deconvolution of the FTIR absorbance bands was added to the caption of Table 3. Furthermore for the N2 physisorption data presented in Table 2 the errors are added to the table.

5) English can be improved, in terms of both grammar and style.

An English check has been conducted and improved where needed.

Round 2

Reviewer 1 Report

The manuscript is ready to be published.

Author Response

-

Reviewer 2 Report

The Authors have addressed all the comments and suggestions raised in the previews round. The paper could be accepted in its present form. I just wonder if the Authors have considered the suggestion to introduce one or two images of the analyzed samples before presenting the spectra. If possible, in my opinion, this would improve the presentation of this research.

Author Response

We thank the reviewer for the remark to add pictures of the samples before analysis.

We have considered this and we do not see any additional benifit to add them since no difference can be observed between the samples before and after treatment. Therefore they have not been added to the revised manuscript.
